# Organelle Stress and Metabolic Derangement in Kidney Disease

**DOI:** 10.3390/ijms23031723

**Published:** 2022-02-02

**Authors:** Reiko Inagi

**Affiliations:** Division of Chronic Kidney Disease (CKD) Pathophysiology, The University of Tokyo Graduate School of Medicine, 7-3-1 Hongo Bunkyo-ku, Tokyo 113-8655, Japan; inagi-r@m.u-tokyo.ac.jp

**Keywords:** metabolomics, acute kidney injury, chronic kidney disease, tubular inflammation, tubular fibrosis, organelle crosstalk, mitochondria, endoplasmic reticulum, primary cilia, lipid metabolism

## Abstract

Advanced multiomics analysis has revealed novel pathophysiological mechanisms in kidney disease. In particular, proteomic and metabolomic analysis shed light on mitochondrial dysfunction (mitochondrial stress) by glycation in diabetic or age-related kidney disease. Further, metabolic damage often results from organelle stress, such as mitochondrial stress and endoplasmic reticulum (ER) stress, as well as interorganelle communication, or “organelle crosstalk”, in various kidney cells. These contribute to progression of the disease phenotype. Aberrant tubular mitochondrial lipid metabolism leads to tubular inflammation and fibrosis. This review article summarizes updated evidence regarding organelle stress, organelle crosstalk, and metabolic derangement in kidney disease.

## 1. Introduction

Significant population growth and aging have contributed to a global increase in chronic kidney disease (CKD). Patients with CKD are susceptible to cardiovascular diseases. Further, complications of CKD, including anemia and mineral-bone disorders, reduce the quality of life of patients. Worldwide, 850 million people are estimated to have some form of kidney disease, making it the 11th leading cause of global mortality. Further, 2.6 million people with end-stage kidney disease receive dialysis or transplantation, and the annual costs of dialysis and transplantation range from 35,000 USD to 100,000 USD per patient. The high costs of these treatment modalities lead to a more serious problem, namely huge gaps between the actual number of patients undergoing kidney replacement therapy, and the estimated number of patients with end-stage kidney disease in low-middle income countries and low-income countries [1]. Analysis of the Global Burden of Disease study data from 1990 to 2016 has revealed increases in the incidence of CKD of 89%, prevalence of CKD of 87%, death due to CKD of 98%, and disability-adjusted-life-years (DALYs) of 62% [2]. One-half of the increased burden of CKD was attributed to diabetes. Thus, it is critical to understand the pathophysiology of kidney disease and develop novel therapeutic approaches.

## 2. Omics Analysis and Kidney Disease

Omics-based medicine in the kidney is an advanced field of research. Genomics and transcriptomics are powerful tools that can reveal changes in those gene expression profiles that contribute to disease phenotypes. In particular, single-cell RNA-sequencing is an advanced technology for finding cell populations linked to disease progression. However, alterations in gene expression pattern are not always enough to allow an understanding of the molecular mechanism of phenotypic changes. Thus, proteomics or metabolomics is a valuable strategy for understanding the relationship between alterations in gene expression and disease phenotype. Omics analyses integrated at the mRNA and protein levels allow us to reveal the novel pathophysiology of kidney disease and subsequently highlight novel therapeutic targets for kidney disease from the point of view of protein homeostasis and metabolism.

### 2.1. Proteomics and Metabolomic Analysis in Glycation Research

Glycation is a non-enzymatic posttranslational protein modification by glucose. It is one of the common reactions related to protein homeostasis (proteostasis) in both food and medical sciences [3,4]. Especially in the medical field, high glucose in diabetes accelerates glycation and, subsequently, affects the structure and function of various proteins in the body [3]. Glycated hemoglobin (HbA1c) is one of the representative glycated proteins generated by high glucose conditions and is a well-known diagnostic biomarker for hyperglycemia. The glycated proteins or glycation precursors are often harmful, both of which are causal factors for various diseases, including kidney disease. Many studies in kidney pathophysiology have demonstrated the pathogenic impact of accumulated advanced glycation endproducts (AGEs), which show the altered protein homeostasis structurally and functionally, in the glomerular lesion associated with mesangial expansion or glomerular endothelial damage. Further, the pathogenic signals via the AGE binding to the receptor for AGE (RAGE) play an important role in the inflammation-mediated endothelial cell injury [3,4]. Nowadays, protein glycation is accelerated by not only hyperglycemia, but also oxidative stress, suggesting the global pathogenic contribution of protein glycation in oxidative stress-related diseases [3,4,5].

Accumulating evidence indicates that updated proteomic analysis unveils the novel pathophysiology of non-enzymatic protein modification by glycation. For example, proteomics analysis of plasma or urine in aging research identified more than 200 age-associated proteins and demonstrated that a substantial group of proteins changed with aging [6]. These include proteins related to the AGE/RAGE pathway. These pathway clusters may be strong candidates for the development of clinical tools for measuring biological aging and predicting changes in healthspan. Based on evidence showing that (1) aging is closely correlated with a decline of kidney function [7] and (2) AGE/RAGE signaling causes various types of kidney cell damage, such as oxidative stress or inflammation [5,8,9], the AGE/RAGE pathway clusters may include novel biomarkers or therapeutic targets for age-related kidney disease.

Recent advances in analytic technology, particularly quantitative liquid chromatography-mass spectrometry (LC-MS), reveal to us details of the metabolome. Accumulated metabolomic data of kidney diseases and subsequent enrichment analyses can elucidate the novel pathophysiological mechanisms of kidney disease. These findings are important to identify novel biomarkers and therapeutic targets based on omics analysis. Prof. Paul Thornally and his colleagues performed urinary metabolomic analysis in 200 human subjects with an early-stage decline in metabolic, vascular, and kidney function and healthy controls, and then applied machine learning to optimize and validate algorithms to discriminate between study groups for potential diagnostic utility [10]. They then demonstrated the benefit of measuring urinary glycated, oxidized, or branched-chain amino acids for early metabolic, vascular, or kidney disease detection as a non-invasive health screen [10]. Prof. Melinda Coughlan’s group performed transcriptomic analysis in the kidney and untargeted metabolome in the cecum from rats fed with a thermally processed diet for 24 weeks to investigate the link between intestinal function and the progression of CKD [11]. The heating-processed diet contained high concentrations of AGEs, including N^ε^-carboxymethyl lysine (CML), N^ε^-carboxyethyl lysine (CEL), and fructosamine. They then revealed how long-term consumption of an AGE-rich diet drives intestinal barrier permeability and risk of CKD. AGE consumption as a component of foods upon heat processing leads to innate immune complement activation, such as increased C5a anaphylatoxin signaling, and subsequent local kidney inflammation associated with a decline of kidney function and tubulointerstitial fibrosis. Importantly, this process was ameliorated by inhibition of complement (C5a) signaling or consumption of a high-fiber resistant starch diet, which has the beneficial effect of maintaining the gut microbiota [9]. These results demonstrate that processed foods with high AGE contents cause the derangement of gut barrier integrity, in turn leading to CKD via complement activation.

### 2.2. Glycation in Mitochondrial Biology

Further, comprehensive proteomic and metabolomic data have provided new insights into mitochondrial alteration as a disease phenotype. The effect of glycation on mitochondrial function has been extensively investigated.

Previous reports have noted that the accumulation of AGEs with aging causes mitochondrial stress in the brain [12]. Aging is known as a risk factor for brain dementia and cognitive decline. Aging increases the accumulation in the body of certain metabolites, such as AGEs or their precursors, highly reactive dicarbonyls that include methylglyoxal (MG), glyoxal (GO), and 3-deoxyglucosone (3-DG). These glycation-related metabolites are often pathogenic factors in the initiation and/or progression of brain damage and neurodegeneration by aging. Akhter et al. demonstrated that arginine- or lysine-derived AGEs, such as CML, CEL, and MG-derived hydroimidazolone-1 (MG-H1), were significantly elevated in the cerebral cortex and hippocampus of aged humans (82 ± 3.41 years) and mice (30 months) [12]. This was associated with declining mitochondrial metabolism and increased production of mitochondrial ROS (oxidative stress). Such increased mitochondrial oxidative stress accelerated AGE formation and accumulation in the brain. Importantly, scavenging mitochondrial reactive oxygen species (ROS) induced by the administration of mitoTEMPO (2-(2,2,6,6-Tetramethylpiperidin-1-oxyl-4-ylamino)-2-oxoethyl)triphenylphosphonium chloride attenuated AGE-induced mitochondrial stress and cognitive impairment, suggesting a link between glycation, mitochondrial metabolism, and brain aging [12]. Prevention of the formation and accumulation of AGEs may be a new therapeutic avenue for combating cognitive decline and mitochondrial stress in age-related neurodegenerative diseases, including Alzheimer’s disease.

Among other findings of note, (1) hepatic RAGE expression increased in aged mice (20 months) and elderly patients with hepatic steatosis, and (2) an age-related increase in RAGE and its ligand AGEs in the liver was correlated with increased hepatic triglyceride (TG) accumulation [13]. The link between RAGE and TG accumulation in the liver may contribute to the progression of non-alcoholic fatty liver disease (NAFLD), which is characterized by increased hepatic TG content. In fact, RAGE expression is negatively correlated with hepatic peroxisome proliferator-activated receptor-α (PPARα) expression, which is a master regulator of mitochondrial lipid metabolism, such as TG retention. The aberrant RAGE/PPARα axis decreases mitochondrial β-oxidation and increases lipid droplet accumulation, leading to hepatic steatosis [13]. These findings demonstrate that the upregulation of RAGE may play a critical role in aging-associated liver steatosis.

AGEs are also involved in the pathogenesis of diabetic nephropathy (DN) via mitochondrial function. Proteomic analysis followed by bioinformatics analysis of the kidney in DN model mice identified the AGE-related proteins as potentially involved in the kidney damage associated with mitochondrial dysfunction [14]. Among these AGE-related proteins, CML decreased the expression of carnitine palmitoyltransferase 2 (CPT2), an enzyme for mitochondrial β-oxidation, which subsequently caused mitochondrial dysfunction and tubular fibrosis. CPT2 overexpression abolished these phenotypic changes, suggesting that CML-induced mitochondrial dysfunction leads to renal fibrosis and DN.

## 3. From Mitochondrial Stress to Interorganelle Communication

Many studies of mitochondrial stress have shed light on how organelle dysfunction, or organelle stress, links to cellular phenotypic changes and subsequent organ damage. In addition, recent research has highlighted that organelles need to communicate with each other to maintain organelle homeostasis and cell fate. Especially, mitochondrial metabolism via interorganelle communication is one of the advanced topics in renal pathophysiology (Table 1). The consensus now exists that the morphology of mitochondrial networks is complex, and that dynamic change in networks is essential for maintaining mitochondrial homeostasis and cell fate. For example, a neuron shows a unique shape characterized by one long cable, the axon. Mitochondria can be conveyed along the axon. The system that conducts this mitochondrial transport dynamically distributes mitochondrial functions, and has proved an attractive field of research [15]. Kinesin family molecules, including Kinesin Family Member 1B (KIF1B), are identified as anterograde motor proteins for the transport of mitochondria [16]. 

### 3.1. Mitochondrial Stress in Kidney Disease

The link between mitochondria and nephrology is also a hot topic. Many studies show dynamic alteration in mitochondrial metabolism, decreased energy production in mitochondria, and abnormal mitochondrial biogenesis in various kidney diseases [17,37]. The kidney is composed of various types of cells that differ both structurally and functionally. From the point of view of mitochondrial morphology, proximal tubular cells are highly mitochondria-rich compared with other kidney cells, such as glomerular cell podocytes. This is reasonable given the high energy demands of proximal tubular cells required for the reabsorption of sodium, glucose, and so on. 

Mitochondrial morphology is regulated by the ongoing fusion and fission of mitochondria, a system required to maintain mitochondrial function. In other words, an imbalance between fusion and fission is a causal factor of cell damage. To address the pathophysiological significance of the mitochondrial fission molecule, dynamin-related protein 1 (Drp1), Prof. Mark Okusa’s group employed proximal tubular-specific Drp1 deficient mice [26]. They found that Drp1 deletion significantly reduced kidney fibrosis caused by ischemia-reperfusion injury, suggesting the pathogenic contribution of Drp1 in tubular damage owing to acute kidney injury (AKI).

In AKI, the pathophysiological role of peroxisome proliferator-activated receptor γ coactivator-1α (PGC1α), a master regulator of mitochondrial biogenesis, has also been extensively investigated. Tubular PGC1α expression is suppressed in AKI, leading to impaired mitochondrial function, such as decreased mitochondrial biogenesis, β-oxidation, and ATP production [18]. 

Mitochondrial dysfunction also contributes to the pathophysiology of CKD [23,24]. Underlying CKD progression is a vicious cycle that leads to increased mitochondrial ROS, decreased ATP production, or alteration of mitochondrial biogenesis and remodeling.

Prof. Farhad Danesh’s group demonstrated that, in diabetic kidney disease (DKD), a major disease of CKD, hyperglycemia triggers mitochondrial fragmentation mediated by the activation of Rho-associated coiled coil-containing protein kinase 1 (ROCK1) in podocytes and endothelial cells [20]. 

It is now the consensus that mitochondrial stress is associated with mitochondrial metabolic alteration, leading to the development and progression of kidney damage, especially kidney inflammation and fibrosis. Shinji Tanaka from our group performed a mass spectrometry-based metabolomic analysis in the kidney of DKD mice (22-week-old BTBR ob/ob diabetic mice) to investigate the pathophysiological role of mitochondrial metabolic alteration [21]. The results showed that these diabetic mice exhibited an abnormal elevation in the renal pools of tricarboxylic acid (TCA) cycle metabolites, which was confirmed by imaging mass spectrometry. Interestingly, when the DKD mice were treated with sodium-glucose cotransporter 2 (SGLT2) inhibitor, this metabolic alteration was almost completely eliminated, associated with the amelioration of kidney damage. Similar results were observed in DKD mice under calorie restriction [21]. These data suggest that the amelioration of mitochondrial stress and metabolic alteration may play a central role in the renoprotective effect of SGLT2 inhibitors or calorie restriction. Mitochondrial stress followed by metabolic damage in DKD was confirmed by further multiomics analysis using transcriptomic and metabolomic analyses. Sho Hasagawa from our group showed the burden of mitochondrial metabolism in the kidney at an early stage in other DKD and CKD model animals, streptozotocin-induced diabetic rats and alloxan-induced diabetic mice, respectively [22]. He also demonstrated that metabolic reprogramming from the TCA cycle to glycolysis is beneficial for preventing disease progression by enarodustat (JTZ-951), an oral hypoxia inducible factor (HIF) stabilizer, suggesting that HIF activation plays a renoprotective role in maintaining mitochondrial homeostasis.

Together with the alteration of mitochondrial homeostasis by glycation, many pathogens, such as high glucose or high glucose-mediated metabolic alteration, impair mitochondrial function quality and result in mitochondrial metabolic derangement. These metabolic changes closely affect the progression of phenotypes shown in various kidney diseases, including CKD and DKD. These findings indicate that mitochondrial quality may be more critical than mitochondrial quantity for maintaining mitochondrial metabolism. In other words, mitochondrial metabolic homeostasis may be a potent strategic point to cure kidney disease and to extend a healthy life span.

### 3.2. Organlle Crosstalk in Kidney Disease

As described in the above section, mitochondrial homeostasis is essential for kidney cell function and morphology, and renal mitochondrial stress or mitochondrial metabolic damage is a common and important factor in various types of kidney cell damage, including inflammation and fibrosis. Advanced mitochondrial research has demonstrated that mitochondrial hemostasis is closely linked to interorganelle communication, namely organelle crosstalk. Mitochondria communicate with other organelles for their function, and the derangement of organelle crosstalk contributes to the kidney disease phenotypes (Figure 1) [38,39].

#### 3.2.1. Mitochondria-Primary Cilia Communication

Mitochondria communicate with primary cilia for tubular cell homeostasis. In autosomal dominant polycystic kidney disease (ADPKD), primary ciliary dysfunction due to mutations in the PKD1 (polycystin-1; PC1) and PKD2 (polycystin-2; PC2) genes, which encode the polycystin 1 and polycystin 2 Ca^2+^ ion channels expressed in the surface of primary cilia, respectively, result in tubular epithelial cell-derived renal cysts. To address the molecular mechanism of cyst formation, multiple groups have investigated and reported mitochondrial functional or morphologic abnormalities in experimental ADPKD models and human patient samples [40,41], including impaired glucose metabolism [28], fatty acid oxidation [29], and disorganized mitochondrial cristae with an altered mitochondrial network [33,34]. Although a functional link between metabolism and the gene products of PKD1 and PKD2 is not yet clear, there have been reports of mitochondrial targeting of a cleavage product of PC1 [34], and PC2 has been proposed to regulate mitochondrial calcium homeostasis [35,36].

Ishimoto, Y. from our group investigated the change in tubular mitochondrial function in ADPKD patients and model animals [33]. The results showed that the primary ciliary dysfunction by PKD1 mutation in the tubules significantly affects PGC1α expression, which is an essential regulator for mitochondrial biogenesis. PKD1-mediated decreases in PGC1α expression exhibited morphological and functional abnormalities, including increased mitochondrial ROS. Moreover, the mitochondrion-specific antioxidant MitoQuinone (MitoQ) reduced such mitochondrial ROS and inhibited cyst epithelial cell proliferation [33]. Collectively, these results indicate that deranged mitochondria-primary cilia communication facilitates cyst formation in ADPKD, highlighting the link between mitochondrial and primary cilia as novel pathophysiology in ADPKD.

It has also been demonstrated that intraflagellar transport 88 (IFT88), a ciliary trafficking protein involved in ciliogenesis, is associated with mitochondrial function [30]. IFT88 expression is decreased in cisplatin-induced AKI and was associated with a shortening of primary cilia length. Interestingly, a decreased IFT88 caused a decrease in mitochondrial lipid metabolism. These findings are consistent with recent papers [31,32], emphasizing the novel regulation mechanism of mitochondrial metabolism by primary cilia.

#### 3.2.2. Mitochondria-Endoplasmic Reticulum (ER) Communication

We have been studying ER dysfunction, namely ER stress, and demonstrated that overwhelming ER stress contributes to glomerular and tubular damage [38,39,42]. Further, our studies also demonstrated the pathophysiological significance of mitochondria-ER crosstalk in tubular damage, including lipid metabolic abnormality, inflammation, and fibrosis. Mitochondria-ER crosstalk may be reasonable given the high energy requirements of the ER for protein synthesis, folding, and subsequent protein homeostasis. 

The unfolded protein response (UPR) pathway is a stress signal that regulates ER homeostasis. Notably, the UPR pathway regulates gene expression related not only to ER homeostasis, but also to lipid metabolism. To determine the impact of the UPR pathway in mitochondrial-ER crosstalk in kidney disease, we assessed tubular fibrosis in mice induced by unilateral ischemia-reperfusion injury and found that ATF 6, which is an ER stress sensor and transcription factor in the UPR pathway, was significantly activated in the damaged tubular cells [27]. Tubular ATF6 activation was associated with lipid droplet accumulation. Notably, ATF6-deficient mice showed the amelioration of tubular fibrosis as well as lipid droplet accumulation. We concluded that, under pathogenic conditions, including tubular fibrosis, tubular ATF6 is activated and subsequently suppresses the promoter activity of PPARα, which is a major regulator of mitochondrial β-oxidation. Decreased PPARα by ATF6 downregulates the downstream gene expression that regulates mitochondrial β-oxidation, and causes lipid accumulation, leading to tubular fibrosis. These data suggest the contribution of crosstalk between ER and mitochondria in lipid metabolism.

We also highlighted another type of mitochondria-ER crosstalk in tubular inflammation. AKI is characterized by mitochondrial dysfunction and activation of the innate immune system. Based on findings that (1) the cyclic GMP-AMP synthase (cGAS) stimulator of interferon genes (STING) pathway detects cytosolic DNA and induces innate immunity and (2) STING is an ER membrane resident molecule, we investigated the role of the cGAS-STING pathway on mitochondrial damage of AKI. The results demonstrated that cGAS-STING activation caused the progression of AKI and that mitochondrial DNA leakage into the cytosol activated cGAS-STING signaling-mediated tubular inflammation [19]. This crosstalk between mitochondria and ER is a novel tubular inflammation pathway that aggravates tubular fibrosis.

#### 3.2.3. Omics-Based Biomarker for Early Detection of DKD

These studies of organelle stress and interorganelle communication enhance our understanding of the novel omics-based pathophysiology of AKI and CKD. They also help identify potent diagnostic or therapeutic targets for kidney disease. In particular, early detection of DKD patients who show fast progression of kidney dysfunction, known as “fast decliners”, is required to lower the risk of a poor prognosis in these patients. We thus performed metabolomic analysis combined with targeted lipidomic analysis to identify the novel mechanism of the rapid progression of DKD. Blood and urine samples from DKD patients at baseline and 10 months were assessed, and the patients were followed for at least 3 years [25]. We evaluated those metabolites that significantly changed in DKD fast decliners and identified lysophosphatidylcholine (LPC), which shows a strong association with DKD prognosis. Of note, an increase in urinary LPC was not correlated with basal kidney function and estimated GFR, but was significantly correlated with the slope of the kidney function decline. These results in human subjects were consistent with the data from DKD rats: urinary LPC level was increased as the disease progressed, and was associated with the accumulation of LPC in the kidney. Imaging mass spectrometry revealed that LPC was mainly accumulated in tubulointerstitium. 

Subcellular fraction analysis in LPC-treated proximal tubular cells showed that the intracellular distribution of LPC is mainly ER in the proximal tubular cells. Importantly, LPC accumulation causes ER stress, estimated by activation of the UPR pathway, and mitochondrial stress, such as decreased fatty acid oxidation associated with morphological abnormalities. These organelle stresses reduced mitochondria-ER contact sites (mitochondria-associated ER membranes: MAM) in proximal tubular cells treated with LPC. Collectively, LPC induced accumulation of tubular lipid droplets via activation of peroxisome proliferator-activated receptor-δ (PPARδ) followed by upregulation of the lipid droplet membrane protein perilipin 2 and decreased autophagic flux, thereby inducing organelle stress and subsequent apoptosis. Thus, LPC may mediate the fast progression of DKD, and serve as a target for novel therapeutic approaches [25].

## 4. Conclusions

In this paper, I summarize how multiomics analysis helps us understand the novel mechanisms of development and progression of disease phenotypes. In particular, disturbance of mitochondrial dysfunction associated with metabolic derangement serves as a common pathogenic pathway in various kidney diseases, such as AKI and CKD. Aberrant organelle crosstalk, including mitochondria-primary cilia and mitochondria-ER, also plays a pathogenic role in tubular inflammation and fibrosis mediated by lipid metabolic alternation (lipotoxicity).

Altogether, mitochondrial metabolism is a promising therapeutic target for various types of kidney diseases. Notably, recently approved new drugs, such as SGLT2 inhibitor and HIF-prolyl hydroxylase (PH) inhibitor, have beneficial renoprotective effects associated with mitochondrial metabolic reprogramming, suggesting the potent strategy for maintaining kidney function via mitochondrial metabolic homeostasis. Pre-clinical studies showed kidney protection by therapies targeting mitochondria in animal models; however, no drug of this kind was validated in a clinical trial until now. Further studies are awaited to develop therapeutic approaches targeting mitochondria in human patients with kidney disease.

## Figures and Tables

**Figure 1 ijms-23-01723-f001:**
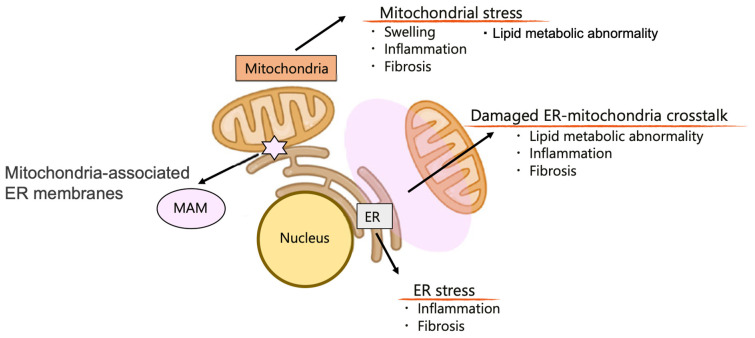
Organelle stress and organelle crosstalk link to lipid metabolic abnormality in the kidney. Omics-based research highlights the importance of metabolic homeostasis in cellular function and subsequent organ function. The metabolic abnormality is caused by organelle dysfunction, such as mitochondrial stress and endoplasmic reticulum (ER) stress, which lead to the disease phenotypes. In the kidney, organelle stress in tubular cells is a risk factor for inflammation and fibrosis mediated by mitochondrial lipid metabolic abnormality (lipotoxicity). Further, interorganelle communication, such as ER-mitochondrial crosstalk or organelle contact site (mitochondria-associated ER membranes: MAM), plays an important role in maintaining organelle homeostasis. Cited by [39] with modification.

**Table 1 ijms-23-01723-t001:** Organelle stress and its phenotypic changes in kidney diseases.

Kidney Disease Pathogenic Phenotypes	Organelle Dysfunction	Organelle Phenotypic Changes	Ref.
Acute Kidney Injury (AKI)			
Ischemia-reperfusion	Mitochondria	↑Mitochondrial fission (Fragmentation) ↓ATP production	[17,18] [17,18]
Cisplatin	Mitochondria	↑Mitochondrial DNA-ER-mediated inflammation	[19]
Chronic Kidney Disease (CKD)			
Diabetic glomerular cell damage	Mitochondria	↑Mitochondrial fission (Fragmentation)	[20]
Diabetic tubular cell damage	Mitochondria ER ER-Mitochondria	Aberrant TCA cycle ↑Mitochondrial fission (Fragmentation) ↑Mitochondrial ROS ↓ATP production ↓Lipid β-oxidation ↑Unfolded protein response (UPR) ↓Organelle contact sites (MAM)	[21,22] [23,24] [23,24] [23,24] [14,25] [25] [25]
AKI (ischemia-reperfusion)-to-CKD transition	Mitochondria ER- Mitochondria	↑Mitochondrial fission (Fragmentation) ↓Lipid β-oxidation (lipotoxicity)	[26] [27]
Autosomal dominant polycystic kidney disease (ADPKD)	Primary cilia-mitochondria	↓Glucose metabolism ↓Lipid β-oxidation ↑Mitochondrial ROS ↑Mitochondrial fission (Fragmentation) Impaired Ca homeostasis	[28] [29,30,31,32] [33,34] [33,34] [35,36]

ER; endoplasmic reticulum, ROS; reactive oxygen species, MAM; mitochondria-associated ER membranes.

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
