# Peer review of "Organelle Stress and Metabolic Derangement in Kidney Disease"

_ijms, 2022, doi:10.3390/ijms23031723_

Round 1
Reviewer 1 Report
This is a very nice review entitle “Organelle stress and metabolic derangement in kidney disease” by Prof. Reiko Inagi sumarazing up-to-dated evidence on metabolic alterations, organelle stress and crosstalk in kidney disease. It is very well written, and summarizes most of the important aspects related to kidney disease and research in the field. I had an enthusiastic reading and I really enjoyed the article. Although, the article needs some complementary information in order to be more complete than it is in the actual version. I´m happy to support the publication of this article if they address the following comments:
Mayor comments:
- The author must include a table that summarizes the most relevant kidney disorders, the type of organelle dysfunction, the metabolic phenotype description, and complementary information considered (references, possible treatment approaches,…).
- Author must include and describe previous and subsequent contributions to the field (including his work as he did) related to “mitochondria and primary cilia communication”. Multiple groups have investigated and reported mitochondrial functional or morphologic abnormalities in experimental ADPKD models and human patient samples (1), including impaired glucose metabolism (2), fatty acid oxidation (4), and disorganized mitochondrial cristae with altered mitochondrial network (5,6). Although a functional link between metabolism and the gene products of PKD1 (polycystin-1; PC1) and PKD2 (polycystin-2; PC2) is not yet clear, there have been reports of mitochondrial targeting of a cleavage product of PC1 (6), and PC2 has been proposed to regulate mitochondrial calcium homeostasis (7,8).
- Takeshi Terabayashi, Luis F. Menezes, Fang Zhou, Hongyi Cai, Peter J. Walter, Hugo M. Garraffo and Gregory G. Germino. Kidney360 October 2021, 2 (10) 1576-1591; DOI: https://doi.org/10.34067/KID.0000962021
- Rowe I, Chiaravalli M, Mannella V, Ulisse V, Quilici G, Pema M, Song XW, Xu H, Mari S, Qian F, Pei Y, Musco G, Boletta A: Defective glucose metabolism in polycystic kidney disease identifies a new therapeutic strategy. Nat Med 19: 488–493, 2013.
- Menezes LF, Germino GG: The pathobiology of polycystic kidney disease from a metabolic viewpoint. Nat Rev Nephrol 15: 735–749, 2019.
- Menezes LF, Lin CC, Zhou F, Germino GG: Fatty acid oxidation is impaired in an orthologous mouse model of autosomal dominant polycystic kidney disease. EBioMedicine 5: 183–192, 2016.
- Ishimoto Y, Inagi R, Yoshihara D, Kugita M, Nagao S, Shimizu A, Takeda N, Wake M, Honda K, Zhou J, Nangaku M: Mitochondrial abnormality facilitates cyst formation in autosomal dominant polycystic kidney disease. Mol Cell Biol 37: e00337-17, 2017.
- Lin CC, Kurashige M, Liu Y, Terabayashi T, Ishimoto Y, Wang T, Choudhary V, Hobbs R, Liu LK, Lee PH, Outeda P, Zhou F, Restifo NP, Watnick T, Kawano H, Horie S, Prinz W, Xu H, Menezes LF, Germino GG: A cleavage product of Polycystin-1 is a mitochondrial matrix protein that affects mitochondria morphology and function when heterologously expressed. Sci Rep 8: 2743, 2018Google Scholar
- Padovano V, Kuo IY, Stavola LK, Aerni HR, Flaherty BJ, Chapin HC, Ma M, Somlo S, Boletta A, Ehrlich BE, Rinehart J, Caplan MJ: The polycystins are modulated by cellular oxygen-sensing pathways and regulate mitochondrial function. Mol Biol Cell 28: 261–269, 2017
- Kuo IY, Brill AL, Lemos FO, Jiang JY, Falcone JL, Kimmerling EP, Cai Y, Dong K, Kaplan DL, Wallace DP, Hofer AM, Ehrlich BE: Polycystin 2 regulates mitochondrial Ca2+ signaling, bioenergetics, and dynamics through mitofusin 2. Sci Signal 12: eaat7397, 2019.
Minor comments:
- Figure 1 is not mentioned in the entire article. I think should be introduce in the text, even if it is a summary of the entire article.
- This is a question for the author. Is the mitochondrial number, apart from the metabolic and structural alterations of the organelle, a phenotype related to kidney disease? If so, would you mind to explain why this happens and the mechanistic consequences of this phenotype?
Author Response
Thank you very much for reviewing this manuscript and for the valuable comments.
This reviewer suggests adding the table, which summarizes the most relevant kidney disorders and the type of organelle dysfunction related to the metabolic phenotype description. I agreed with the suggestion and added the table for more easy understanding (cited on page 4).
Another comment is that the author must include and describe previous and subsequent contributions to the field related to “mitochondria and primary cilia communication”. I missed describing the previously published important works, thus mentioned it as the reviewer suggested on page 5 (the 4th paragraph). Thank you very much again for pointing out the important points.
“To address the molecular mechanism of cyst formation, multiple groups have investigated and reported mitochondrial functional or morphologic abnormalities in experimental ADPKD models and human patient samples [28,29], including impaired glucose metabolism [30], fatty acid oxidation [31], and disorganized mitochondrial cristae with an altered mitochondrial network [32,33]. Although a functional link between metabolism and the gene products of PKD1 and PKD2 is not yet clear, there have been reports of mitochondrial targeting of a cleavage product of PC1 [33], and PC2 has been proposed to regulate mitochondrial calcium homeostasis [34,35].”
Minor comments:
- As the reviewer suggests introducing figure 1 in the text, I cited it on page 5 (line 256).
- Regarding the question “Is the mitochondrial number, apart from the metabolic and structural alterations of the organelle, a phenotype related to kidney disease?”, I added the sentences to respond to it as follows: (on page 5, the 2nd paragraph).
“Together with the alteration of mitochondrial homeostasis by glycation, many pathogens, such as high glucose or high glucose-mediated metabolic alteration, impair mitochondrial function quality and result in mitochondrial metabolic derangement. These metabolic changes closely affect the progression of phenotypes shown in various kidney diseases, including CKD and DKD. These findings indicate that mitochondrial quality may be more critical than mitochondrial quantity for maintaining mitochondrial metabolism. In other words, mitochondrial metabolic homeostasis may be a potent strategic point to cure kidney disease and to extend a healthy life span.”
Reviewer 2 Report
In this, Dr. Inagi, a renowned export in the field of AGEs, organelle stress and pathophysiology of kidney disease, discussed recent advancement of the field. The manuscript is well-written and holds great value of research updates, however, I think it fell “short” - many citations are review articles and not well-around. It will be great if citations include work from other research groups also. (Out of 32 citations, 9 are self-citation.)
Recommendations to expand this manuscript a bit --
It will be great to give a brief historical overview on “AGE Glycation in kidney-specific pathology”;
Conclusion and Perspectives –any pharmacological agents to reduce organelle stress, or to enhance energy metabolism in kidney disease.
A typo-
P5L217- it is ADPKD (not ADPKC)
Author Response
Thank you very much for reviewing this manuscript and for the valuable comments.
This reviewer suggests expanding the manuscript and adding a brief historical overview on “AGE Glycation in kidney-specific pathology”. According to the suggestions, I added the following sentences on page 2:
“Glycation is a non-enzymatic posttranslational protein modification by glucose. And glycation is one of the common reactions related to protein homeostasis (proteostasis) in both food and medical sciences [3, 4]. Especially in the medical field, high glucose in diabetes accelerates glycation and subsequently affects the structure and function of various proteins in the body [3]. Glycated hemoglobin (HbA1c) is one of the representative glycated proteins generated by high glucose conditions and is a well-known diagnostic biomarker for hyperglycemia. The glycated proteins or glycation precursors are often harmful, both of which are causal factors for various diseases, including kidney disease. Many studies in kidney pathophysiology have demonstrated the pathogenic impact of accumulated advanced glycation endproducts (AGEs), which show the altered protein homeostasis structurally and functionally, in the glomerular lesion associated with mesangial expansion or glomerular endothelial damage. And the pathogenic signals via the AGE binding to the receptor for AGE (RAGE) play an important role in the inflammation-mediated endothelial cell injury [3, 4]. Nowadays, protein glycation is accelerated by not only hyperglycemia but also oxidative stress, suggesting the global pathogenic contribution of protein glycation in oxidative stress-related diseases [3-5].”
Another comment is that in the session of Conclusion and Perspectives any pharmacological agents to reduce organelle stress, or to enhance energy metabolism in kidney disease should be mentioned. I appreciate the valuable comment for improving this manuscript and added the following sentences on page 7.
“Altogether, mitochondrial metabolism is a promising therapeutic target for various types of kidney diseases. Notably, recently approved new drugs, such as SGLT2 inhibitor and HIF-prolyl hydroxylase (PH) inhibitor, have beneficial renoprotective effects associated with mitochondrial metabolic reprogramming, suggesting the potent strategy for maintaining kidney function via mitochondrial metabolic homeostasis. Pre-clinical studies showed kidney protection by therapies targeting mitochondria in animal models; however, no drug of this kind was validated in a clinical trial until now. Further studies are awaited to develop therapeutic approaches targeting mitochondria in human patients with kidney disease.”
I also correct the typo error on page 5, line 281.
Thank you very much again for the helpful comments for revising the manuscript.